# Multi-Omics Analysis Reveals Up-Regulation of APR Signaling, LXR/RXR and FXR/RXR Activation Pathways in Holstein Dairy Cows Exposed to High-Altitude Hypoxia

**DOI:** 10.3390/ani9070406

**Published:** 2019-07-01

**Authors:** Zhiwei Kong, Chuanshe Zhou, Liang Chen, Ao Ren, Dongjie Zhang, Zhuzha Basang, Zhiliang Tan, Jinhe Kang, Bin Li

**Affiliations:** 1CAS Key Laboratory for Agro-Ecological Processes in Subtropical Region, National Engineering Laboratory for Pollution Control and Waste Utilization in Livestock and Poultry Production, South-Central Experimental Station of Animal Nutrition and Feed Science in Ministry of Agriculture, Hunan Provincial Engineering Research Center for Healthy Livestock and Poultry Production, Institute of Subtropical Agriculture, The Chinese Academy of Sciences, Changsha 410125, China; 2College of Resource and Environment, University of the Chinese Academy of Sciences, Beijing 100049, China; 3Hunan Co-Innovation Center of Safety Animal Production, CICSAP, Changsha 410128, China; 4College of Animal Science and Technology, Hunan Agricultural University, Changsha 410128, China; 5Institute of Animal Husbandry, Heilongjiang Academy of Agricultural Sciences, Harbin 150028, China; 6Institute of Animal Husbandry and Veterinary, Tibet Autonomous Regional Academy of Agricultural Sciences, State Key Laboratory of Hulless Barley and Yak Germplasm Resources and Genetic Improvement, Lhasa 850000, China

**Keywords:** plasma proteomic, microRNA analysis, Holstein dairy cows, high-altitude hypoxia

## Abstract

**Simple Summary:**

Blood has been widely collected and analyzed for diagnosing and monitoring diseases in human beings and animals. A range of plasma proteins and peptides were set as biomarkers for pathological and physiological status. Previous researchers have explored how humans, pigs, dogs, and horses adapt to hypoxia at high altitudes. Additionally, the mechanism of hypoxia adaptation in human, mice, and shrimp was studied by proteomics. However, information on the adaptation mechanism of Holstein cows introduced to high altitudes is limited. The present study was conducted to the adaptation mechanism of Holstein dairy cows to high-altitude hypoxia by miRNA microarray analysis and the isobaric tags for relative and absolute quantitation (iTRAQ) iTRAQ technology. Based on the obtained results, Holstein dairy cows transported to Nyingchi may adapt to the high-altitude hypoxia through regulation of inflammatory homeostasis by up-regulating the acute phase response (APR) APR and activation of the liver X receptor/retinoid X receptor (LXR/RXR)LXR/RXR and farnesoid X receptor/ retinoid X receptor (FXR/RXR) FXR/RXR pathways.

**Abstract:**

Changes in the environment such as high-altitude hypoxia (HAH) high-altitude hypoxia can lead to adaptive changes in the blood system of mammals. However, there is limited information about the adaptation of Holstein dairy cows introduced to high-altitude areas. This study used 12 multiparous Holstein dairy cows (600 ± 55 kg, average three years old) exposed to HAH conditions in Nyingchi of Tibet (altitude 3000 m) and HAH-free conditions in Shenyang (altitude 50 m). The miRNA microarray analysis and iTRAQ proteomics approach (accepted as more suitable for accurate and comprehensive prediction of miRNA targets) were applied to explore the differences in the plasma proteomic and miRNA profiles in Holstein dairy cows. A total of 70 differential miRNAs (54 up-regulated, Fold change (FC) FC > 2, and 16 down-regulated, FC < 0.5) and 226 differential proteins (132 up-regulated, FC > 1.2, and 94 down-regulated, FC < 0.8) were found in the HAH-stressed group compared with the HAH-free group. Integrative analysis of proteomic and miRNA profiles demonstrated the biological processes associated with differential proteins were the immune response, complement activation, protein activation, and lipid transport. The integrative analysis of canonical pathways were most prominently associated with the APR signaling (z = 1.604), and LXR/RXR activation (z = 0.365), and FXR/RXR activation (z = 0.446) pathways. The current results indicated that Holstein dairy cows exposed to HAH could adapt to high-altitude hypoxia by up-regulating the APR, activating the LXR/RXR and FXE/RXR pathways.

## 1. Introduction

High-altitude (defined as 8000 ft above sea level) hypoxia (HAH) continuously affect physical and mental performances of people and animals [1]. HAH affects the immune system, makes the human body more susceptible to various infectious and autoimmune diseases [2], and decreases the production of cytokines [3]. Additionally, hypoxia is associated with the homeostasis and metabolic rate of adult tissues [4]. With the adaptation of hypoxia, the level and activity of glycogen synthase increases *in vivo* [5]. Hypoxia increases fatty acid metabolism, and reduces fat synthesis and storage [6]. Previous researchers have explored how humans, pigs, dogs, and horses adapt to high-altitude hypoxia [7]. Additionally, the proteomics approach was applied to study the adaptive mechanism to hypoxia in human [8], mouse [9], and oriental river prawn [10]. However, there is limited information about the adaptation of Holstein dairy cows introduced to the high-altitude areas. 

Holstein dairy cows are the most widely distributed and the largest number of rearing in the world. They are famous for good production performance, especially high milk yield [11]. In terms of production performance, the average slaughter rate can reach over 50%, and the net meat rate is above 40% [12]. Compared with Jersey cattle, Holstein dairy cows have significantly higher milk production, peak day and adult equivalent [13]. The practices in the Qinghai province of China have proven that, after more than ten years of directional cultivation, “Qinghai Holstein dairy cows” with strong resistance to crude feed, strong disease resistance, high milk yield and stable genetic performance are cultivated [14]. Therefore, great interest is focused on the adaptation of Holstein dairy cows to high-altitude regions.

Previous studies report that miRNAs regulate various cellular and biological processes [15]. Studies have focused on the role of miRNAs in regulating the adaptation to hypoxia in endothelial cell [16], cardiomyocyte [17], and lung [18], including miR-19a-5p, miR-532-5p, and miR-150. However, the role of miRNAs in the adaptation to high-altitude hypoxia in blood is poorly understood, particularly miRNA-regulated molecular networks. Proteomics is supposed to be able to identify miRNA targets [19] and explore the molecular pathogenesis of miRNA-regulated diseases [20]. In order to better understand the adaptation of Holstein dairy cows to high-altitude hypoxia, miRNA microarray analysis and tandem mass spectrometry (QSTAR) QSTAR Elite liquid chromatography with tandem mass spectrometry (LC-MS/MS), coupled with the isobaric tags for relative and absolute quantitation (iTRAQ) technology, were applied to detect the difference in plasma of Holstein dairy cows originated from low- and high-altitude regions. 

## 2. Materials and Methods 

This experiment was approved by the Animal Care Committee of Institute of Subtropical Agriculture, The Chinese Academy of Sciences, Changsha, China, with protocol ISA-201705.

### 2.1. Animals and Experimental Design

Twelve multiparous Holstein dairy cows (600 ± 55 kg, average 3 years old, dry period) were blocked into 6 blocks based on body weight. Within blocks, Holstein dairy cows were randomly assigned to 1 of the 2 groups (6 cows per group). One group was fed in Nyingchi of Tibet (HA group, altitude 3000 m), while the other group was fed in Shenyang (SL group, altitude 50 m). The experiment was conducted in autumn for 30 days. The environment factors are shown in Table 1. 

The basal diet (shown in Table 2) was formulated based on the *Feeding Standards of Dairy Cattle in China* to satisfy the nutritional requirements of Holstein dairy cows [21]. Holstein dairy cows in both groups were fed the same Total Mixed Rations (TMR) diets ad libitum. 

### 2.2. Sample Preparation

Pre-prandial blood samples were collected from the jugular vein of all experimental animals on the last day of the experiment by heparinized vacuum tubes. Plasma was obtained by centrifugation at 3000× g for 15 min at 4 °C and stored (−80 °C) until iTRAQ proteomics and miRNA microarray analysis. Part of the pre-prandial blood samples were collected by serum tubes, centrifuged at 3000× g for 10 mins at 4 °C and stored (−40 °C) until analyzed for immune index.

### 2.3. Analysis of the Blood Immune Index

Cow ELISA kits (Invitrogen, Carlsbad, CA, USA) were applied to determine the levels of Interleukin (IL)-2 and IL-6, while a Cow Radioimmunoassay kit (North Institute of Biotechnology, Beijing, China) was selected to measure the level of tumor necrosis factor-α (TNF-α).

### 2.4. Total RNA Extraction and miRNA High-Throughput Sequencing

TRIzol reagent (Invitrogen, Carlsbad, CA, USA) was selected to extract total RNA of the plasma. The purification, quality, and integrity of total RNA was conducted by the mirVanaTM miRNA Isolation kit (Ambion, Austin, TX, USA), Qubit^®^ 2.0 Fluorometer (Invitrogen, Carlsbad, CA, USA) and Agilent 2100 Bioanalyzer (Agilent Technologies, Santa Clara, CA, USA), respectively. The Illumina small RNA sample preparation protocol was selected by Shanghai Biotechnology Corporation to be the basis of library preparation and Illumina sequencing [23]. In brief, firstly the purified small RNA molecules were ligated to a 5’adaptor and a 3’adaptor by T4 RNA ligase. Secondly, SuperScript II Reverse Transcription Kit (Invitrogen, Carlsbad, CA, USA) was applied to reverse transcribe the adapter-ligated small RNAs into cDNA. Then an Agilent 2100 Bioanalyzer (Agilent Technologies, Santa Clara, CA, USA) was used to detect the quality and concentrations of purified PCR products. In the end, a Qubit Fluorometer (Invitrogen, Carlsbad, CA, USA) was selected to quantify the purified cDNA libraries, while the Illumina Genome Analyzer IIx (Illumina, San Diego, CA, USA) was applied to cluster and analyze the 36 nt single-end sequencing.

### 2.5. Analysis of miRNA High-Throughput Sequencing Data

According to previous research methods [24], the raw data obtained from Illumina sequencing was analyzed. By filtering out the adapter sequences and low-quality reads, unique reads of 18–25 nt were selected for further analysis. The CLC Main workbench 4.9 software (Illumina, San Diego, CA, USA) was applied to identify the known miRNAs. Based on the previously published methods [25], the expression of transcript per million was obtained. The miRNAs with ratio value of ≥ 2 or ≤ 0.5 in the HAH group were identified as the significantly changed miRNAs. Adjusted t-test method was applied to analyze whether the difference of the selected miRNAs was significant between HA and SL groups. The miRNAs with significant difference (0.01 < *p* < 0.05), or extremely significant difference (*p* < 0.01) between HA and SL groups, were referred to as HAH-associated miRNAs. 

### 2.6. Protein Extraction, iTRAQ Labeling, and Strong Cation Exchange (SCX) Chromatography

The albumin in the blood was removed by using a Seppro Bovine Serum Albumin depletion kit (Sigma Aldrich, St. Louis, MO, USA) [26]. According to a procedure described previously [27], total proteins of the plasma were extracted. In brief, sample were extracted with Lysis buffer 3 (8 M Urea, 40 mM Tris-HCl or TEAB, pH 8.5) containing 1mM phenylmethylsulfonyl fluoride (PMSF) and 2 mM Ethylene Diamine Tetraacetic Acid (EDTA). Subsequently, the samples were incubated with 55 mM iodoacetamide for 45 min for alkylation. The supernatant containing proteins was quantified by Bradford after centrifuge with 25,000 g × 20 mins at 4 °C. Protein denaturation, reduction, and alkylation were performed based on a former research procedure [28]. Strong Cation Exchange (SCX) chromatography was performed on the basis of previous research methods [29].

### 2.7. LC-MS/MS Analysis for Protein Identification and Quantitation

According to the previous research methods [30], LC-MS/MS was applied to identify the proteins by Q-Exactive mass spectrometer (Thermo Fisher Scientific, Waltham, MA, USA). Analyzing the peptide data, Proteome Discover 1.4 software (Thermo Fisher Scientific, Waltham, MA, USA) was selected to process the raw mass data. Mascot 2.2 (Matrix Science, London, UK) was applied to search the SwissProt database (uniprot-Bovine, 2016_11_13). Based on the previously published methods [31], Proteome Discoverer 1.4 software was applied to determine the quantification of iTRAQ labeled unique peptides, while reporter ion intensities was used as the criterion of the level of the expressive abundance. According to the methods previously published [32], a significance score (*p*-value) for log protein ratios was calculated. Based on the above analysis, a cutoff point at 50% variation and fold change (FC) > 1.2 or < 0.8 within an error factor < 2 was applied to identify differentially expressed proteins (DEPs) between the two groups.

### 2.8. Validation of miRNA and iTRAQ Data

To validate the reliability of miRNA high-throughput sequencing data, six miRNAs related to HAH adaptation (miR-143, miR-181a, miR-199a, miR-19b, miR-17-5p, and let-7a-5p) were selected for qRT-PCR validation. The primer sequences (including specific stem-loop reverse transcription primers and qRT-PCR primers) are provided in Appendix A. cDNA was synthesized using the Avian Myeloblastosis Virus (AMV) reverse transcription kit (Promega, Madison, WI, USA). The qRT-PCR was performed using SYBR Green Supermix (BioRad, Hercules, CA, USA). 

Commercially available ELISA kits (bovine 1B glycoprotein, serum amyloid A4, apolipoprotein D, intermediate-globulin inhibitor H3, CUSABIO, city, China) (company, city, state abbrev if USA, country) were applied to determine the expression levels of Alpha-1B-glycoprotein (A1BG), Serum amyloid A-4 protein (SAA4), Apolipoprotein D (APOD), Alpha -trypsin inhibitor heavy chain H3 (ITIH3)A1BG, SAA4, APOD, and ITIH3 in plasma according to the manufacturer’s instructions. A microplate reader (Multiskan Spectrum Microplate Reader, Thermo Fisher Scientific, Waltham, MA, USA) was used to examine the absorbances at 450 nm.

### 2.9. Bioinformatics Analysis 

In order to better understand the biological functions of miRNAs and proteins identified in our miRNA high-throughput sequencing and quantitative iTRAQ data, Blast2Go 4.0.7 software (Illumina, San Diego, CA, USA) [33] was applied to classify the biological processes of gene ontology (GO) of the differentially expressed miRNAs (DEMs) and proteins according to the procedure described previously [34]. 

Additionally, according to the previous research methods [35], Ingenuity Pathway Analysis (IPA) software (version 9.0; Qiagen Bioinformatics, Redwood City, CA, USA) was applied to identify associated biological functions and predominant canonical pathways between the DEMs and DEPs. Based on the procedure described previously [36], the significance of the association between the DEPs and canonical pathways was measured. In this study, pathways with *p*-values < 0.05 were supposed to be significant in the HA group.

### 2.10. Statistical Analysis

MiRNAs with differential abundance between the two groups were considered with FC ≥ 2 or ≤ 0.5 and *p*-value < 0.05. When FC ≥ 1.2 or ≤ 0.5 and *p*-value ≤ 0.05, proteins were supposed to be differentially expressed.

Additionally, an independent-samples t-test method was applied to detect the difference in serum cytokines levels, while adjusted t-test method was used to identify the *p*-value of DEMs and DEPs.

## 3. Results

### 3.1. Serum Cytokines Levels

As is shown in Table 3, when compared with HAH-free Holstein dairy cows (SL group), the levels of cytokines interleukin-2 (IL-2), IL-6, and TNF-α were decreased significantly (*p* < 0.05) in HAH-stressed Holstein dairy cows (HA group). 

### 3.2. Differentially Expressed miRNAs and Protein Profiles

In this study, the FastQC software was applied to evaluate the data quality, and effectively filter and control the data according to the quality control results. The sequencing depth in this experiment was 100×, 1 μg RNA was input for library preparation, and finally 31,747,628 reads were obtained on sequencing. A total of 468 known miRNAs were identified in the plasma of all Holstein dairy cows selected using miRNA microarray analysis. Among them, 16 miRNAs were down-regulated with FC < 0.5, while 54 miRNAs were up-regulated with FC > 2 in HAH-stressed Holstein dairy cows with expression abundances > 20 and priority verified *p* < 0.05, compared with HAH-free Holstein dairy cows (Appendix A).

iTRAQ-based LC-MS/MS quantitative proteomic approach was employed to assess proteome changes in plasma of Holstein dairy cows exposed to HAH. A total of 1313 proteins were identified. Based on a previous description [37], 226 DEPs (132 up-regulations with FC > 1.2 and 94 down-regulations with FC < 0.8) were obtained between the HA and SL groups. The characteristic and description findings of DEPs were shown in Appendix A. 

Moreover, based on the above datasets, an integrative analysis was conducted by using the IPA software, and an interaction network of miRNAs and their targets are shown in Figure 1. In the integrative network, four distinctive genes including 40S ribosomal protein S15 (RPS15), Protein argonaute-2 (AGO2), Ribonuclease 3 (DROSHA) and Baculoviral IAP repeat-containing protein 5 (BIRC5) RPS15, AGO2, DROSHA and BIRC5 were targeted by more than three miRNAs. Further analysis showed that the miRNAs and their targets in the interaction network were mainly involved in the acute phase response (APR) signaling, liver X receptor/retinoid X receptor (LXR/RXR) and farnesoid X receptor/retinoid X receptor (FXR/RXR) activation pathways. Therefore, we inferred that six miRNAs and 38 proteins synergistically regulated the adaptation to HAH in Holstein dairy cows, mainly through the above mentioned signaling pathways.

In the present work, six miRNAs associated with APR signaling, and LXR/RXR and FXR/RXR activation pathways were significantly changed and used for subsequent analysis, including up-regulated miRNAs (let-7a-5p, miR-17-5p, miR-199a-3p, and miR-19b) and down-regulated miRNAs (miR-143 and miR-181a). The results of relative miRNA expression showed that compared with SL, the level of let-7a-5p, miR-17-5p, miR-199a-3p, and miR-19b were significantly increased in HA, while the level of miR-143 and miR-181a were significantly decreased in HA (shown in Figure 2A). 

Among the differential proteins associated with HAH adaptation, four proteins, including Alpha-1B-glycoprotein (A1BG) (No.PRC6415), Serum amyloid A-4 protein (SAA4) (No.YZ-S0267J), Apolipoprotein D (APOD) (No.1529820332), and Alpha -trypsin inhibitor heavy chain H3 (ITIH3) (No.XY-SJH-4652), were used for ELISA detection (Figure 2B) to validate the reliability of the iTRAQ results. The results showed that the concentration of A1BG and ITIH3 were significantly increase in HA, while the concentration of SAA4 and APOD were significantly decreased in HA, which was consistent with the results of iTRAQ (Figure 2B).

### 3.3. Gene Ontology, Function, and Pathway Analysis

GO enrichment was performed for functional analysis of the 70 differential miRNAs and 226 differential proteins. The results were categorized into biological processes, cellular components, and molecular functions. The top twenty categories of the GO enrichment analysis of differential miRNAs and proteins, shown in Figure 3, confirmed that the differential miRNAs were overwhelmingly associated with localization, transmission, and system organization, while the differential proteins were overwhelmingly associated with immune response, complement activation, protein activation, and lipid transport in biological processes. 

The canonical pathways of integrative analysis obtained from the IPA software are shown in Figure 4. In the pathway analysis, we found 16 statistically significant enriched canonical pathways associated with the 38 deregulated plasma proteins of Holstein dairy cows (Table 4). Among these, the top three according to the *p*-value were the APR signaling (20 proteins: APOA1, SAA4, APOA2, F1MKS5, F1N076, G3N1U4, F1N0R5, HEMO, FETUA, A2AP, TRFE, PEDF, Q0VC51, ITIH3, HPT, Q5GN72, F1MNW4, K4JF16, TTHY, and A0A140T897), LXR/RXR activation (10 proteins: Q2KIW1, F1N5M2, F1MNN7, APOD, Q2KIW4, F1N3Q7, APOC4, A8DBT6, APOC2, and Q3ZBS7), and FXR/RXR activation (eight proteins: APOC3, A0A140T881, E1BJF9, F1MWI1, A1AT, A1BG, RET4, and FETUB). The top three –log(*p*-value) corresponding to APR signaling, LXR/RXR activation, and FXR/RXR activation in plasma showed a z-score > 0 (z = 1.604, z = 0.365, and z = 0.446, respectively).

## 4. Discussion

Previous studies show that high-altitude hypoxia affects the functioning of the immune system, making the human body more susceptible to various infectious and autoimmune diseases. In this study, compared with SL group, compared with SL group HAH-free Holstein dairy cows, the level of cytokines IL-2, IL-6, and TNF-α decreased in Holstein dairy cows exposed to high-altitude hypoxia (HA group), which was consistent with the results described previously [38].

In the interaction network, we found three miRNAs involved in the APR signaling. Let-7 is a kind of miRNA found across species [39]. A previous study reports that the pro-inflammatory and apoptotic effects of let-7a are achieved through its target of protein-tyrosine-phosphatase MKP1 [40]. In addition, Chen and his colleagues find that let-7 regulates the anti-inflammatory responses by targeting TLR/NF-kB signaling pathways [41]. MiR-181a suppresses immune inflammatory responses in various kinds of cells in vitro [42,43]. Studies also show that increased miR-181a expression promotes the homeostatic response to inflammatory stimuli by TLR-4 pathway activation in vivo [44]. In addition, Gantier and his colleagues discover that miR-19b contributes to inflammatory activation by targeting NF-kB signaling [45]. In the present work, the above three miRNAs were all up-regulated, which contributed to induce the APR to promote the homeostatic response to inflammatory stimuli. 

A study reports that miR-143 is confirmed to be closely associated with stable blood glucose levels and insulin sensitivity [46]. Additionally, as one of the members of glycolysis regulation, miR-143 affects progress of glycolysis by regulating the target gene HK2 [47]. A previous study finds that miR-17 impairs glucose metabolism through expression inhibition of facilitated glucose transporter member 4 (GLUT4) [48]. Wang reports that glucose metabolic pathways can be down-regulated by targeting let-7a [49]. In the present work, these three above miRNAs were all found associated with LXR/RXR activation in the interaction network, and the miR-17 and let-7a were up-regulated, while the miR-143 was down-regulated, which was consistent with the results described previously that under the condition of hypoxia, in order to adapt to the environment, the aerobic metabolism pathway of glucose was blocked, and the anaerobic metabolism pathway was strengthened, thus affecting the organism blood glucose levels [50]. 

It is reported that in ruminant cells, miR-181a negatively regulates lipid synthesis by targeting ACSL1, while it regulates lipid metabolism by targeting the cytosolic isocitrate dehydrogenase enzyme and controls adipogenesis by targeting tumor necrosis factor-α in non-ruminant cells [51]. Previous studies report that miR-199a-3p regulates the lipid metabolism, promotes adipocyte proliferation, and represses adipocyte differentiation [52]. In this paper, the above two miRNAs were all up-regulated, which would lead to increased lipid metabolism, and this was consistent with the result from this experiment that the FXR/RXR activation pathway was activated. 

From the results of integrative analysis by IPA software, we obtained 20 differential proteins associated with the APR signaling. The APR is involved in multiple biological processes protecting the organism from local or systemic stimuli during inflammatory processes [53]. In these processes, tissue damage induced by inflammation is usually accompanied by the release of pro-inflammatory cytokines, which contributes to regulate the synthesis of the acute phase reactants in liver [54]. The APR is a systemic immune-mediated response associated with increased serum amyloid A, ceruloplasmin, haptoglobulin, α1-acid glycoprotein, and α-2-macroglobulin, and decreased transferrin, apolipoprotein A, and albumin [55]. Haptoglobin has the functions of anti-inflammation, antioxidant, detoxification, and angiogenesis, and it is usually up-regulated in response to inflammatory stress [56]. Ceruloplasmin (Cp) and Serotransferrin (Trf) levels are increased in response to inflammation, infection, malnutrition, and perturbations in iron metabolism. Hypoxia induces Cp synthesis [57] at a post-transcriptional level, while inflammation regulates Trf and Cp synthesis at a transcriptional level [58]. In inflammatory states, α1-acid glycoprotein levels [59] and albumin synthesis [60] has been observed to increase. In the present work, the expression of transferrin, ceruloplasmin, haptoglobulin, α1-acid glycoprotein, albumin, and α-2-macroglobulin were up-regulated, which led to the adaptation to high-altitude hypoxia by responding to the inflammatory stimuli.

The integrative analysis also identified the significant enrichment of the LXR/RXR and FXR/RXR activation pathways, which were both predicted as activated with z-scores of 0.365 and 0.446, respectively. Hypoxia can stimulate lipolysis and inhibit the uptake of free fatty acids (FFA) [61], then decrease the fatty acid biosynthesis [62], which may partly result from citrate reduction. The proteins (APOC2, LCAT, and APOC4) involved in positive regulation of fatty acid biosynthetic process were down-regulated in this paper, which is consistent with the results described above. Furthermore, the conversion of glucose into citrate is prohibited under hypoxia due to the inhibition of the tricarboxylic acid (TCA) cycle [63], which may lead to the down-regulation of the RBP4 protein involved in gluconeogenesis in the present work. The RXR is a nuclear hormone receptor of the retinoid receptor family, and it usually works with receptors such as LXR and FXR [64]. These receptors are often found in biological and pathological pathways involved in glucose and lipid homeostasis and inflammatory responses [65]. In general, when LXR is combined with oxysterols, the sterol regulatory element binding protein 1c (SREBP-1c) will be activated [66]. The activation of SREBP-1c leads to the regulation of genes associated with lipogenesis and lipoprotein metabolism [67], which are up-regulated in Holstein dairy cows in this work. In addition, LXR activates reverse cholesterol transport and down-regulates various pro-inflammatory transcription genes [68], while FXR activates cholesterol homeostasis, triacylglyceride metabolism, and suppresses inflammatory processes [65]. The anti-inflammatory properties of both pathways suggest that activation of LXR/RXR and FXR/RXR pathways reduces the inflammatory responses in Holstein dairy cows exposed to HAH. 

## 5. Conclusions

As a whole, since all the different biological pathways involved this integrative analysis were related to inflammation, we pointed to this event as crucial in the adaption to high-altitude hypoxia in Holstein dairy cows. The activation of LXR/RXR and FXR/RXR pathways could attempt to alleviate the inflammation. Therefore, we deduce that Holstein dairy cows in Nyingchi may adapt to the exposure to HAH through regulation of inflammatory homeostasis by up-regulating the APR and activation of LXR/RXR and FXR/RXR pathways. 

## Figures and Tables

**Figure 1 animals-09-00406-f001:**
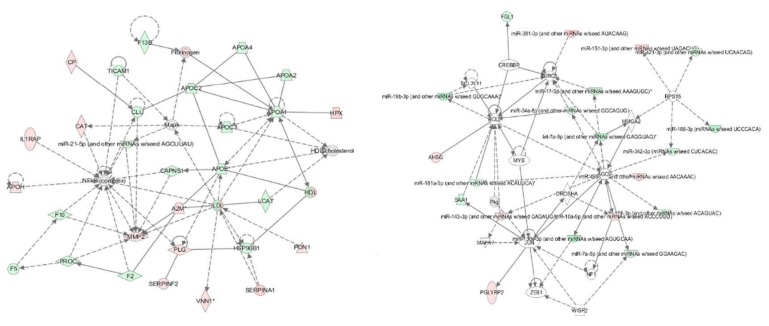
An interaction network of the miRNAs and their targets in the plasma of HAH-stressed Holstein dairy cows. IPA software (Illumina, San Diego, CA, USA) was used to generate the interaction network of miRNAs and their targets. Each node indicated a single protein or miRNA. Red color indicates up-regulated miRNAs, and green color indicates down-regulated proteins. Lines and arrows indicate connections between two objects; broken lines indicate indirect relationships, and solid lines indicate direct relationships.

**Figure 2 animals-09-00406-f002:**
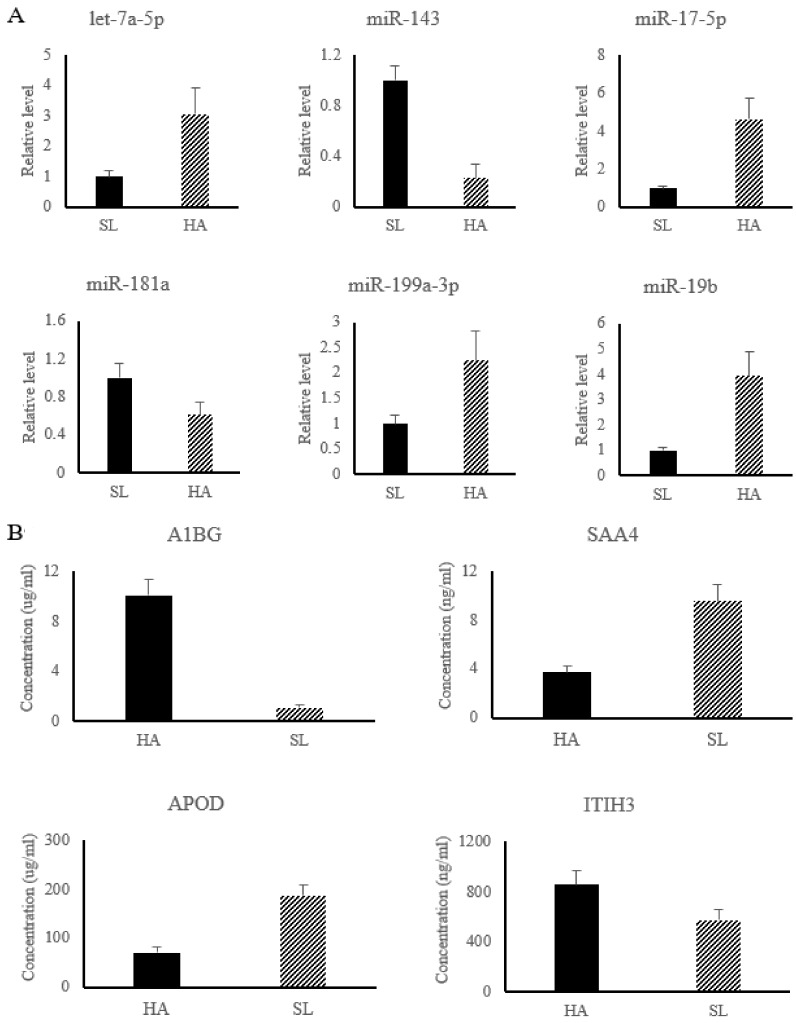
The relative level of differential miRNAs were detected by qRT-PCR (**A**), while the concentrations of differential proteins were determined by ELISA kit (**B**). HA group, Nyingchi, 3000 m above sea level; SL group, Shenyang, 50 m above sea level. Note: A1BG, Alpha-1B-glycoprotein; SAA4, Serum amyloid A-4 protein; APOD, Apolipoprotein D; ITIH3, Alpha -trypsin inhibitor heavy chain H3.

**Figure 3 animals-09-00406-f003:**
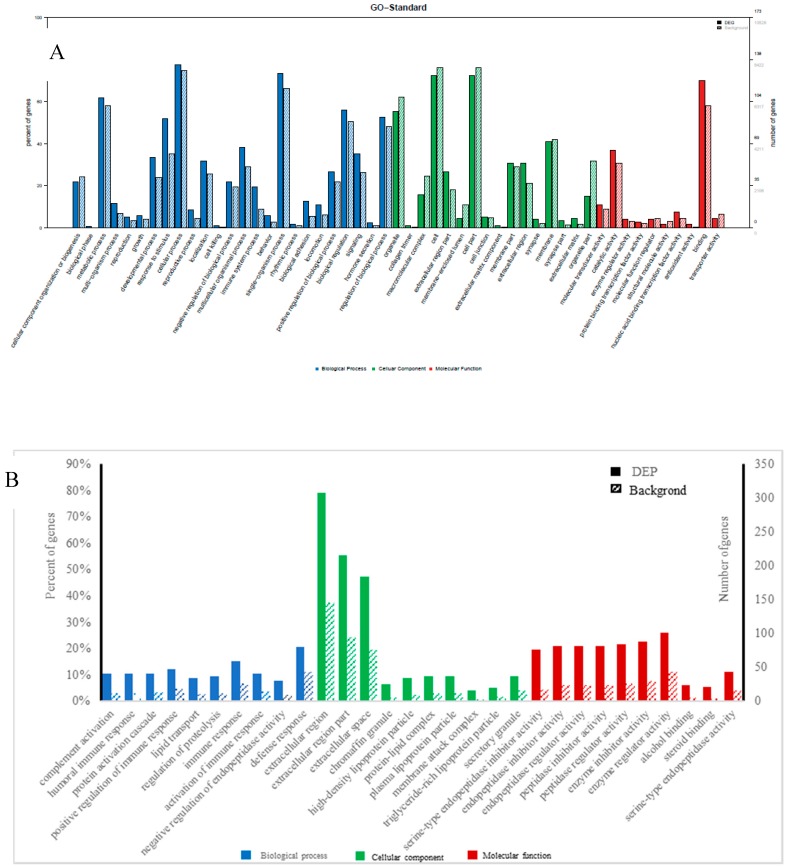
Gene ontology (GO) classification diagram of differentially expressed miRNAs (**A**) and proteins (**B**), Note: The horizontal coordinate is the classification name of the GO annotation information, the left vertical coordinate is the percentage of the number of proteins/genes, and the right vertical coordinate is the number of proteins/genes. The solid line spline is the GO annotation information of the protein/the target gene of miRNA with differential expression, and the dotted line spline is the GO annotation information of the protein/gene in the 3’-noncoding region (3UTR) (or mRNA) database.

**Figure 4 animals-09-00406-f004:**
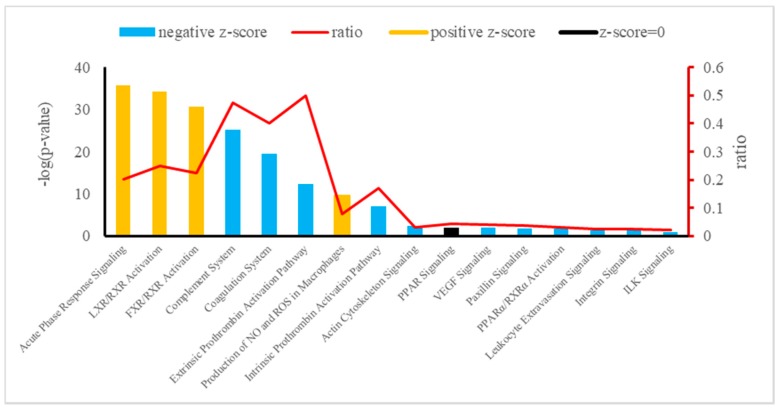
Canonical pathways enriched in Holstein cows exposed to HAH by integrative analysis. Orange or blue colors mean positive or negative z-score, respectively. LXR/RXR = liver X receptor/retinoid X receptor; FXR/RXR = farnesoid X receptor/retinoid X receptor; NO = nitric oxide; ROS = reactive oxygen species; PPAR = peroxisome proliferator-activated receptors; VEGF = vascular endothelial growth factor; PPARα/ RXRα = peroxisome proliferator-activated α receptor/retinoid X α receptor; ILK = integrin-linked kinase.

**Table 1 animals-09-00406-t001:** Environmental factors in both groups.

Item	Temperature	Humidity	Oxygen Content (%)
HA	10.6 ± 1.6	42.4 ± 3.6	13.86 ± 1.22
SL	11.8 ± 1.2	48.5 ± 2.7	21 ± 0.87

Note: HA, means Holstein cows fed in Nyingchi of Tibet; SL group, means Holstein cows fed in Shenyang.

**Table 2 animals-09-00406-t002:** Ingredient compositions and nutrition levels of the diets (% of DM).

Items	Contents (%)
Diet compositions	
Chinese leymus	37.5
Corn silage	22.5
Corn	15.2
Wheat bran	5.3
Soybean meal	9.2
DDGS)	8.4
Calcium hydrophosphate	1.4
Premix ^1^	0.5
Nutrient compositions	
CP	13.1
NDF	39.6
Ca	0.6
P	0.4
NEL ^2^, MJ/kg DM	5.4

^1^ Premix, one kilogram of premix contained mixed vitamins 800,000 IU; Fe, 1500 mg; Cu, 1000 mg; Zn, 11,000 mg; Mn, 3500 mg; Se, 80 mg; I, 200 mg; Co, 50 mg. ^2^ NEL was calculated according to the results described previously [22]. Note: DDGS, Distiller’s dried grain with solubles; CP, Crude protein; NDF, Neutral detergent fibers; NEL, Net energy for lactating cow; DM, Dry matter.

**Table 3 animals-09-00406-t003:** Effects of high-altitude hypoxia (HAH) on the levels of serum cytokines in HAH-stressed Holstein dairy cows (n = 6).

Items ^1^	Group ^2^		*p*-Value
	SL	HA	
IL-2 (ng/L)	0.32 ± 0.028	0.18 ± 0.021	0.002
IL-6 (ng/L)	0.17 ± 0.009	0.11 ± 0.013	0.001
TNF-α (ng/L)	1.54 ± 0.095	1.12 ± 0.049	0.001

^1^ IL, interleukin; TNF-α, tumor necrosis factor-α. ^2^ SL, Holstein dairy cows fed for 30 d in Shenyang, Liaoning (50 m above sea level; SL group); HA, Holstein dairy cows fed for 30 d in Nyingchi, Tibet (3000 m above sea level; HA group). Values represent mean ± SD.

**Table 4 animals-09-00406-t004:** Differentially expressed proteins in the plasma of Holstein dairy cows exposed to HAH associated with APR, LXR/RXR, and FXR/RXR activation.

Functional Classification	Names of Proteins	HA/SL: Ratio ^1^	*p*-Value	NCBInr Accession ^2^
Acute Phase Response	ceruloplasmin precursor (CP)	1.72	0.001	gi|375065868
	Serpin A3-6 (SERPINA3)	4.32	0.009	gi|296475221
	von Willebrand factor (VWF)	1.40	0.008	gi|328887902
	Hemopexin (HPX)	7.34	0.001	gi|77736171
	Alpha-2-HS-glycoprotein (AHSG)	4.03	0.017	gi|27806751
	Alpha-2-antiplasmin (SERPINF2)	1.95	0.001	gi|27807209
	Serotransferrin (TF)	4.22	0.016	gi|2501351
	Pigment epithelium-derived factor (SERPINF1)	5.01	0.011	gi|27806487
	Interleukin 1 receptor protein (IL1RAP)	3.66	0.028	gi|115495597
	Alpha -trypsin inhibitor heavy chain H3 (ITIH3)	1.5	0.035	gi|156120445
	Haptoglobin (HP)	1.77	0.006	gi|94966763
	Alpha-1-acid glycoprotein (ORM1)	9.05	0.014	gi|122697593
	Alpha -trypsin inhibitor heavy chain H2 (ITIH2)	2.73	0.039	gi|296481520
	Alpha -2-macroglobulin variant 23 (A2M)	6.12	0.001	gi|408689609
	Transthyretin (TTR)	5.75	0.021	gi|27806789
	Serum albumin (ALB)	8.97	0.001	gi|30794280
	Apolipoprotein A-I preproprotein (APOA1)	0.32	0.002	gi|75832056
	Serum amyloid A-4 protein (SAA4)	0.39	0.005	gi|94966809
	Apolipoprotein A-II (APOA2)	0.65	0.003	gi|114052298
	Histidine-rich glycoprotein (HRG)	0.61	0.001	gi|27806875
LXR/RXR Activation	Paraoxonase 1 (PON1)	1.50	0.004	gi|114053183
	Vitamin D-binding protein (GC)	4.73	0.004	gi|296486435
	Lipopolysaccharide-binding protein (LBP)	0.23	0.013	gi|296481091
	Apolipoprotein D (APOD)	0.38	0.001	gi|115494984
	Lecithin-cholesterol acyltransferase (LCAT)	0.41	0.043	gi|114051546
	Apolipoprotein A-IV (APOA4)	0.23	0.008	gi|296480272
	Apolipoprotein C-IV (APOC4)	0.36	0.005	gi|77736596
	Monocyte differentiation antigen CD14 (CD14)	0.62	0.022	gi|157703516
	Apolipoprotein C-II (APOC2)	0.37	0.001	gi|156139070
	vitronectin precursor (VTN)	0.66	0.002	gi|78045497
FXR/RXR Activation	Alpha-1-antiproteinase (SERPINA1)	7.21	0.023	gi|27806941
	Alpha-1B-glycoprotein (A1BG)	9.05	0.003	gi|114053019
	Retinol-binding protein 4 (RBP4)	7.02	0.045	gi|164420709
	Fetuin-B (FETUB)	8.89	0.002	gi|77735387
	Apolipoprotein C-III (APOC3)	0.45	0.001	gi|47564119
	Apolipoprotein E (APOE)	0.33	0.041	gi|27806739
	Clusterin (CLU)	0.41	0.006	gi|47522770
	Serum amyloid A protein (SAA1)	0.59	0.005	gi|296471870

^1^ Ratio of HA group (Nyingchi, Tibet; 3000 m above sea level) to SL group (Shenyang, Liaoning; 50 m above sea level) cattle. ^2^ National Center for Biotechnology Information.

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
