# Peer review of "Multi-Omics Analysis Reveals Up-Regulation of APR Signaling, LXR/RXR and FXR/RXR Activation Pathways in Holstein Dairy Cows Exposed to High-Altitude Hypoxia"

_animals, 2019, doi:10.3390/ani9070406_

Round 1
Reviewer 1 Report
In this study entitled “Multi-omics analysis reveals up-regulation of APR signaling, LXR/RXR and FXR/RXR activation pathways in Holstein dairy cows exposed to high altitude hypoxia” Kong et al., profiles the adaptation mechanism of Holstein cows at high altitudes by evaluating their molecular profiles using next generation sequencing approaches. Author’s reports interesting changes in molecular profile between high altitude and low altitude exposed cattle experiments. They further report the possibility of APR signaling pathway and other activation pathways playing role in adaptation phenomenon in cows at high altitudes.
This study is significant in terms of identifying the underlying factors HAH adaptation in cattle, however authors should address following concerns:
Specific Points:
1. In Result section, authors should provide details of RNA sequencing. What was the sequencing depth? How many reads were obtained upon sequencing? What was the input RNA amount for library preparation?
2. Author should also include in detail the sequencing QC steps they undertook before analyzing the data.
3. Authors should provide the Bianalyzer traces of RNA seq libraries as supplementary information.
4. Authors have labelled Fig. 2 as Fig. 4. Same should be corrected in the manuscript.
5. Authors have repeatedly used “former research methods” in the manuscript, authors are suggested to be consistent and use “previously published methods” instead of “former research methods”.
6. Regarding Fig. 2 barplots authors are suggested to be consistent with bar type pattern. Use same pattern (solid) for HA in both A and B figure panel.
Author Response
Response:
Thank you for the comments proposed by both reviewers. When I revised the manuscript according to the proposed comments, I found that the professional knowledge and writing proficiency had been improved. This is very important for the revision and the next writing.
Thank you once again.
The responses are as follow:
1. In Result section, authors should provide details of RNA sequencing. What was the sequencing depth? How many reads were obtained upon sequencing? What was the input RNA amount for library preparation?
AU: we have revised and supplied the information required in the paper.
2. Author should also include in detail the sequencing QC steps they undertook before analyzing the data.
AU: we have revised in the paper.
3. Authors should provide the Bianalyzer traces of RNA seq libraries as supplementary information.
AU: we have supplied this supplementary information in a separate file and hand it along with the revised manuscript.
4. Authors have labelled Fig. 2 as Fig. 4. Same should be corrected in the manuscript.
AU: we have revised in the paper.
5. Authors have repeatedly used “former research methods” in the manuscript, authors are suggested to be consistent and use “previously published methods” instead of “former research methods”.
AU: we have revised in the paper.
6. Regarding Fig. 2 barplots authors are suggested to be consistent with bar type pattern. Use same pattern (solid) for HA in both A and B figure panel.
AU: we have revised in the paper.
Reviewer 2 Report
This study investigates the adaptation of Holstein cows to high attitude hypoxia by multi analyses of blood samples. This reviewer recognizes that this manuscript includes novel points, and also finds few points better to be considered. Please see a few concerns below.
Major concerns:
1. (Line 111-116) The authors used 12 cows in this study, and divided them into two groups, one is high attitude and the other is not high attitude. How is the number of cows in these two groups? And the authors recognize the difference of the data of two groups as the difference of oxygen conditions (hypoxia or not). Is it true? This is because cows received effects not only by high attitude hypoxia, but also temperature, humidity, wind, buildings and many other factors. Thus, it is better to show at first some evidence (in their data) that the difference of the data of two groups at least includes the difference of oxygen conditions, and then to analyze the whole data like in the present form of manuscript.
Minor concerns:
2. Appearance of abbreviations are not consistent. Full term should be appeared one time with its abbreviation, and the abbreviation is used thereafter. It should be checked throughout the MS. For example, HAH, APR, GO, etc..
3. (Line 116) “the experiment was conducted in autumn for 30 days”: Is this only in one year?
4. (Line 133-135) Information of regents are limited. Almost all the catalogue numbers are not cited. For example, the information of antibodies is important for readers.
5. (Line 227, Table 2) Upper case letter “3” is not needed.
6. (Line 376-385) Why this part is blue color?
Author Response
Response:
Thank you for the comments proposed by both reviewers. When I revised the manuscript according to the proposed comments, I found that the professional knowledge and writing proficiency had been improved. This is very important for the revision and the next writing.
Thank you once again.
The responses are as follow:
Major concerns:
1. (Line 111-116) The authors used 12 cows in this study, and divided them into two groups, one is high attitude and the other is not high attitude. How is the number of cows in these two groups? And the authors recognize the difference of the data of two groups as the difference of oxygen conditions (hypoxia or not). Is it true? This is because cows received effects not only by high attitude hypoxia, but also temperature, humidity, wind, buildings and many other factors. Thus, it is better to show at first some evidence (in their data) that the difference of the data of two groups at least includes the difference of oxygen conditions, and then to analyze the whole data like in the present form of manuscript.
AU: The 12 cows were randomly assigned to 1 of the 2 groups (6 cows per group), and we have revised in the paper. The table in the following was the average value of some index, thus we pay more attention on the hypoxia effects.
Table 1 Environmental factors in both groups
Temperature | Humidity | Oxygen content (%) | |
HA | 10.6±1.6 | 42.4±3.6 | 13.86 ±1.22 |
SL | 11.8±1.2 | 48.5±2.7 | 21± 0.87 |
Minor concerns:
2. Appearance of abbreviations are not consistent. Full term should be appeared one time with its abbreviation, and the abbreviation is used thereafter. It should be checked throughout the MS. For example, HAH, APR, GO, etc.
AU: we have revised in the paper
3. (Line 116) “the experiment was conducted in autumn for 30 days”: Is this only in one year?
AU: The experiment was conducted only in one year. We will consider doing this experiment in a longer period in the future.
4. (Line 133-135) Information of regents are limited. Almost all the catalogue numbers are not cited. For example, the information of antibodies is important for readers.
AU: we have considered your suggestion and supplied the information required in the paper.
5. (Line 227, Table 2) Upper case letter “3” is not needed.
AU: we have revised in the paper.
6. (Line 376-385) Why this part is blue color?
AU: It may be the version problem, or the submission is wrong, the current version shows the normal color.